# Application of an Artificial Neural Network to Automate the Measurement of Kinematic Characteristics of Punches in Boxing

Ilshat Khasanshin 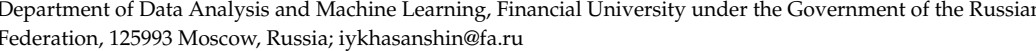

Department of Data Analysis and Machine Learning, Financial University under the Government of the Russian Federation, 125993 Moscow, Russia; iykhasanshin@fa.ru

**Abstract:** This work aimed to study the automation of measuring the speed of punches of boxers during shadow boxing using inertial measurement units (IMUs) based on an artificial neural network (ANN). In boxing, for the effective development of an athlete, constant control of the punch speed is required. However, even when using modern means of measuring kinematic parameters, it is necessary to record the circumstances under which the punch was performed: The type of punch (jab, cross, hook, or uppercut) and the type of activity (shadow boxing, single punch, or series of punches). Therefore, to eliminate errors and accelerate the process, that is, automate measurements, the use of an ANN in the form of a multilayer perceptron (MLP) is proposed. During the experiments, IMUs were installed on the boxers' wrists. The input parameters of the ANN were the absolute acceleration and angular velocity. The experiment was conducted for three groups of boxers with different levels of training. The developed model showed a high level of punch recognition for all groups, and it can be concluded that the use of the ANN significantly accelerates the collection of data on the kinetic characteristics of boxers' punches and allows this process to be automated.

**Keywords:** artificial neural network; inertial measurement units; boxing; punch recognition; multilayer perceptron; automation of measuring of boxers' kinematic parameters

## 1. Introduction

The punch speed is one of the most important characteristics of a boxer; the ability to punch quickly and unexpectedly almost always decides the course of the fight and brings victory. Therefore, coaches work tirelessly on the development of the speed qualities of athletes and this has led to a high activity of scientists in this field.

In order for the speed of a boxer's punch to develop, it is necessary not only to conduct the right training efforts, but also to constantly monitor the progress. Therefore, the development of the athlete takes place under the close attention of the coach, since without tracking the dynamics of change, it is impossible to understand whether the choice of training strategy and tactics is correct. Constant control is associated with a large number of measurements of the speed and acceleration of punches. This is a great burden on coaches, who train dozens of athletes. These measurements are especially difficult to carry out in the conditions of a real fight or shadow boxing, although it is these parameters of punches that are most interesting for coaches. In addition, the notation of the speed parameters of punches in such a fast process as a real fight in boxing is associated with human errors [1].

Therefore, the aim of this study was to develop an automated system for collecting and analyzing data on the speed parameters of punches in a shadow fight.

Movements in boxing are complex and, therefore, as can be seen in the works of other authors in this field, mainly single punches that were struck outside of a real fight are analyzed. Therefore, the aim was chosen considering the next step in development—the recognition of punches in shadow boxing. Moreover, we selected a narrow aim for shadow

boxing—to only make single punches. The next aim of future research will be to study the series of punches in shadow boxing, followed by the recognition of punches in a real fight.

To analyze the kinematic properties and technique of boxers' punches in modern conditions, various methods and means are used, including inertial measurement units (IMUs), video motion capture, exoskeleton, and robotic solutions [2]. Complex solutions can often be found when methods are applied simultaneously, for example, [3].

At the same time, one should not lose sight of the real needs of coaches. Therefore, IMUs were used to measure the velocity parameters of punches, since other solutions are expensive and require complex mathematical models for recognizing punches and a high computing power, since they are associated with video processing [1].

The statics, kinematics, and dynamics of human movements are the result of a complex interaction of muscles, tendons, and bones. Human movement has a complex structure. The kinematic structure includes spatial, temporal, rhythmic, and phase characteristics of movement. The dynamic structure of movement is represented by power, energy, and inertial parameters. The anatomical structure of movement is determined by the human musculoskeletal system. The control structure is controlled by sensory and psychological characteristics. To build a mathematical model describing human movement, taking into account the interaction of all these structures, which are also largely individual for different people, is very difficult. A large number of tools and methods are currently well-developed and used to build a biomechanical model, including tensometry, accelerometers and gyroscopes, and electromyography, which provide data on the speed, acceleration, and rotation angles of body segments; electrical activity of muscles; support reaction forces; and movement trajectory. However, the integration of this mass of data into a single picture of movement still faces many difficulties [4].

An alternative method for analyzing human movements is the use of machine learning methods. The artificial neural network (ANN) provides many opportunities for data analysis in sports and improving the efficiency of athletes' movements.

In this work, the input data for the ANN were obtained using IMUs, which included a gyroscope and an accelerometer. IMUs were attached to both of the boxer's wrists. The use of the ANN presupposes the presence of a large amount of data, but modern technologies offer researchers miniature sensors, wireless data transmitters, and fast computer processing of a large amount of information for analyzing movements. To analyze the data, we used the ANN in the form of a multilayer perceptron.

The athletes were instructed to strike as fast and hard as possible, as in a boxing fight. In addition, the instructions included punching in a variety of ways, including to the head, to the body, backward, and along with forward movement, realizing the principle of "repetition without repetition", which was formulated by the great Russian neurophysiologist and researcher of sports biomechanics N. Bernstein. Bernstein suggested that for the construction of movements of varying complexity, commands are given at different levels (hierarchical levels) of the nervous system. When automating movements, control functions are transferred to a lower (unconscious) level [5]. In [6], it is stated that "Seven decades ago, the Russian physiologist and movement scientist Nikolai A. Bernstein proposed a hierarchical model to explain the construction of movements. In his model, the levels of the hierarchy share a common language (i.e., they are commensurate) and perform complementing functions to bring about dexterous movements". Therefore, the repetition-no-repetition principle was applied to increase both the diversity of the ANN training dataset and the training effect.

There are many scientific studies devoted to individual aspects of the kinematics and dynamics of movements in boxing; however, the application of neural networks to the study of dynamics of the striking technique in martial arts has not yet been sufficiently developed. For example, in a systematic literature review [1], it is shown that out of 52 publications, only three works are devoted to the analysis of data on punches in combat sports based on machine learning. Moreover, in the work [7], it is shown that fifteen years

before 2019, out of 36 publications devoted to the use of IMUs in martial arts, only four works use machine learning.

## 2. Related Works

An interesting and comprehensive review of the implementation of machine learning approaches for motion recognition in various sports is provided in [1]. The aim of this study was to review works on the recognition of sports movements using deep and machine learning based on two tools: IMUs and video capture of movements.

In total, 52 works were reviewed, and IMUs and video motion capture were used in approximately half of the publications. Twelve studies used a deep learning method as a form of Convolutional Neural Network algorithm and one study adopted a Long Short Term Memory architecture in their model. This work indicates that "The implementation of automated detection and recognition of sport-specific movements overcomes the limitations associated with manual performance analysis methods." This highlights the relevance of our work.

Additionally, in [1], it is noted that the advantages of wearable IMUs include the fact that they are wireless, lightweight, and self-contained in operation. However, for computer vision applications, automated approaches to motion recognition require several pre-processing steps, including athlete detection and tracking, time-clipping, and the recognition of targeted actions, which depend on the sport and type of footage captured. As a result of the review, the authors could not state a clear preference for using IMUs or video motion capture, considering that "In addressing the research questions, both IMUs and computer vision have demonstrated capacity in improving the information gained from sport movement and skill recognition for performance analysis".

Only three works in the review [1] are devoted to the application of machine learning models in martial arts.

In [8], an overview of the main areas of research on the use of machine learning in the field of sports was carried out, including motion analysis, performance evaluation, sports data capture, the generation of eating plans, training planning, strategic planning, the prediction of results/patterns, sports data analytics, and decision-making support. The work [9] provides the main methods of data analysis employed in sports based on the methods of machine learning, consisting of Recurrent Neural Networks (RNN), Long Short-Term Memory (LSTM), and Convolutional Neural Networks (CNN).

A model obtained with the help of an ANN is essentially a "black box"; however, this model is very useful for the practical purposes of automating sports tests and analyzing sports equipment, together with the dynamics and kinematics of movements. In this sense, the authors of [1] note that the use of an ANN to automate the collection of kinematic data in sports has great prospects. For example, in [10], the authors argue that biomechanical feedback is key to improving the athletic performance. In addition, they suggest using IMUs and deep machine learning to implement biofeedback.

The works [11,12] use video motion capture. Therefore, the use of machine learning in motion recognition in martial arts is mostly based on video motion capture.

In [11], "the task is to detect boxers' upper body parts to classify the six basic punches; straight, hook and uppercut (for both lead and rear hands), from overhead depth images". The analysis of movements was carried out on the basis of one method: The Support Vector Machine (SVM) and Breiman Random Forest. Four experiments were conducted, and each dataset consisted of 300–400 punches. As stated in the work, "Multi-class SVM classifiers and a Breiman Random Forest are utilized for punch classification. Experimental results illustrate the accuracy of 97.3% on previously seen athletes, and 96.2% on unseen athletes". The limitations that are characteristic of video capture are that the camera is located on top of the athletes and boxers punch single blows and cannot move freely.

In [12], Kinect for Windows v2 as a data capturing device was applied. Punch and kick classification was carried out with Gestures Description Language and the Hidden Markov Model (HMM) classifier. Six people participated in the study. The paper states,

"the dataset consists of 10 classes of actions and included dynamic actions of stands, kicks and blocking techniques. Total number of samples was 1236". As a result of the research, "the recognition rate of our methodology differs between karate techniques and is in the range of $81 \pm 15\%$ even to 100%". The classification was carried out using only one method based on a small dataset in comparison with our work and obtained high results. It should also be noted that the recognition of strikes and blocks was carried out under certain conditions, far from those of a real fight. These conditions included single strikes and the athletes having to stand in a certain place. These limitations are typical for the recognition conditions when using video capture. This is in contrast to our work, in which athletes could move freely.

In [13], the posture-based graph method was applied to analyze the movements, while the shadow boxing motions of the boxer were captured using an optical motion capture system. Visualization of movements was one of the main objectives of the study. Classification was also performed using only one method—HMM. Here, a very interesting approach to movement recognition is applied—tracking poses when moving. As the authors write in [13], "the posture-based graph focuses on evaluating the common postures that are used to start and end actions. In such a graph, the nodes represent similar postures and the edges represent similar actions". This approach expands the recognition capabilities and eliminates the problem of "binding" to the place where the athlete should be. However, this method is also very complex and expensive compared to the one described in our work, although it allows more diverse effects to be achieved.

Through video capture, an analysis of the kinematics of punches and kicks in various types of combat sports was carried out in [14]. No punch recognition or classification was performed.

IMUs are actively used in martial arts. In [7], a review of the use of IMUs for performance analysis in combat sports is presented. The authors analyzed 36 works in which IMUs were used to determine parameters such as the strike quality, strike classification, strike frequency, automatic scoring, movement speed (footwork), and power output. Data analysis was carried out based on the following methods: The Classification Tree (CT); Convolution Neural Network (CNN); K-Nearest Neighbors (KNN); Support Vector Machine (SVM); Dynamic Time Warping (DTW); and Radial Basis Function Neural Network (RNN). The installation of IMUs on all body segments was rarely used (only in four cases); IMUs were mainly installed on the wrists or shins. Accelerometers were used in all works. In 13 works, an additional gyroscope was used, and in six works, a magnetometer was also used. Machine learning-based data analysis was applied in four papers. That is, the majority of scientists only installed the IMUs on the impact segments of the body and did not use a magnetometer, which is often included in IMUs. Approximately half of the studies used IMUs with an accelerometer with a measurement limit of $\pm 16$ g or lower. For this type of research, this is a sufficient level.

The authors of [3], when studying the recognition of movements in fencing, applied an integrated approach—they simultaneously applied IMUs (accelerometer data) and video capture of movements. In the work, preliminary multistage processing of the input data was carried out to supply them to the input of the multilayer perceptron (MLP). The paper shows that before data were sent to the MLP, they were processed using SVM, which gave good results. At the same time, when only processing SVM, the results on the classification of movements were worse than the final results, after processing in the MLP.

The authors of [15] conducted a study of punch velocities by installing an accelerometer on the wrist. The study involved 16 people. The acceleration was averaged over the standard deviation, and the mean integral speed was then found from this calculated value of acceleration. It can be noted that the averaging also included negative acceleration, which occurs in the initial phase of the motion. The authors did not use punch recognition and research automation; they even had to use video viewing of the strikes to determine the phases of the strikes. The maximum punch speed was 16 m/s.

The most similar research in the field and research methods is the work [16]. In this work, a study on the recognition of boxers' punches was conducted based on various machine learning methods: "Six machine learning models were evaluated, this included the logistic regression (LR), linear support vector machine (LSVM), Gaussian rbf support vector machine (GSVM), multi-layer perceptron neural network (MLP-NN), random forest (RF) and gradient boosting (XGB) algorithms". The input parameters of the machine learning models were the IMU data, which were installed on boxers in two ways: "(configuration 1—inertial sensor worn on both wrists; configuration 2—inertial sensor worn on both wrists and third thoracic vertebrae". The authors concluded, "For sensor configuration 1, a support vector machine (SVM) model with a Gaussian rbf kernel performed the best (accuracy = 0.96), for sensor configuration 2, a multi-layered perceptron neural network (MLP-NN) model performed the best (accuracy = 0.98)."

## 3. Materials and Methods

### 3.1. Design of Experiments

The general scheme of the experiment is shown in Figure 1. IMUs were fixed on both wrists of the boxer, including a gyroscope and an accelerometer, which allowed the rotational and translational movements of the hands to be tracked. Along with the IMUs, wireless transmitters were installed on the wrists, which, via a Bluetooth channel, transmitted data to a computer for analysis using the ANN. Boxers punched in shadow boxing mode. The data of the measuring modules began to be transmitted to the computer with a frequency of 1 millisecond after reaching a certain threshold of the absolute value of acceleration, thus determining the moment when the punch was made. With this solution, the data transmission channel was less loaded.

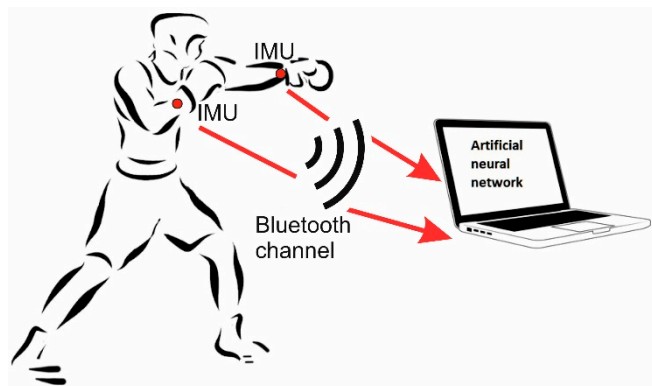

**Figure 1.** Design of the experiment.

The value of absolute acceleration after which the program decided that this was the beginning of the punch was 2.5 m/s$^2$. Of course, this method of determining the beginning of the punch has a number of disadvantages. The main drawback is that the boxer can make quick hand movements that the system can mistake for a punch. However, this disadvantage is compensated for by the fact that the classification in the ANN included the category "movements without punch", which eliminated incorrect cases.

The boxers wore standard sportswear and 10-ounce boxing gloves.

After the ANN model was trained, it was saved in a file with the extension .h5, together with the topology of the network (weights and bias). This file was then converted and embedded in the microcontroller using the X-CUBE-AI code generator [17]. Classification based on the trained data from the experiments was carried out in a microcontroller.

### 3.2. Measuring Modules

The IMUs were fixed on the wrist in such a way that the *y*-axis of the accelerometer was located along the direction of the punch, and the *x*-axis was located along the thumb

line (see Figure 2). The directions of the positive direction of rotation of the gyroscope are shown in Figure 2.

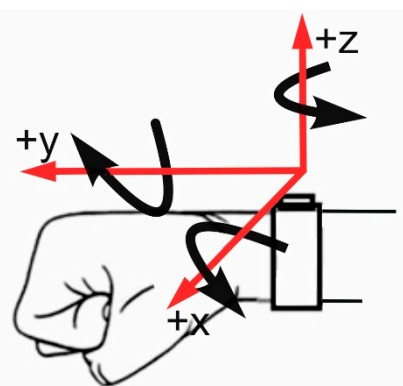

**Figure 2.** Hand with the inertial measurement unit (IMU) on the wrist. The positive directions of the accelerometer axes are shown, as well as the positive directions of the angular rate.

The measuring modules were digital, and were controlled and transmitted data using a digital protocol. In our case, the i2c interface was used (the sampling frequency was 400 kHz). The technical characteristics of the measuring module were as follows:

Accelerometer—sensitivity = $\pm16$ g and nonlinearity = 0.2%, and

gyroscope—sensitivity = $\pm2000°$/s and nonlinearity = 0.5%.

The weight of the box with the microcontroller, IMU, and Bluetooth module was 35 g.

The measuring modules were connected to microcontroller devices. Since accelerometers and gyroscopes are characterized by high levels of data noise, the microcontroller processed the data of the accelerometer and gyroscope using the Kalman filter. The microcontroller also controlled the transfer of data from the IMU to the computer using a Bluetooth module (Figure 3).

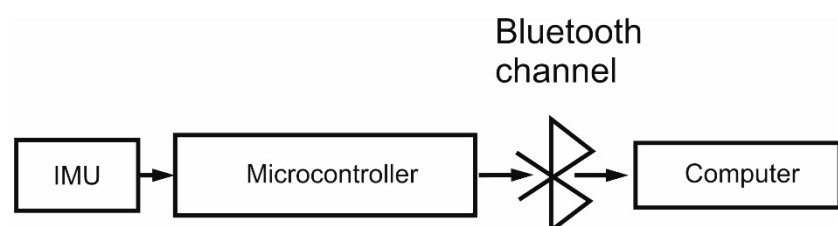

**Figure 3.** Schematic diagram of the device operation.

The following data were transmitted to the computer: The absolute acceleration ($a_{abs}$) and angular rate ($g_{abs}$) every 1 millisecond for the duration of 300 milliseconds. The stm32f407 microcontroller was used [18] with the possibility of implementing a stored ANN model.

### 3.3. Neural Network Architecture

To analyze the kinematic data of punches, the ANN model in the form of a multilayer perceptron was used [19,20].

MLP is a feedforward artificial neural network model. This is one of the oldest methods of deep learning, and a special case of the Rosenblatt perceptron, which was developed in 1957. When training, the MLP uses a supervised learning algorithm of back propagation of error [19,20]. MLPs have shown the ability to find approximate solutions for extremely complex problems. In particular, they are a universal function approximator, so they are successfully used in the construction of regression models. Since classification can be considered a special case of regression, when the output variable is categorical, classifiers can be built on the basis of MLP.

In our work, the use of MLP is due to the fact that this model solves classification problems very well. In [16], where the recognition of boxers' punches was performed by various machine learning methods, MLP proved to be the most effective algorithm.

The disadvantages of MLP are its low learning rate, as well as the need to select the structure of the neural network for a specific task. Furthermore, the choice of topology is often based on heuristic methods, as described in the book [19]: "The number of hidden neurons should be between the size of the input layer and the size of the output layer. The number of hidden neurons should be 2/3 the size of the input layer, plus the size of the output layer. The number of hidden neurons should be less than twice the size of the input layer". However, these rules, or others, can only act as a starting point for the empirical design of the MLP architecture. For example, in the aforementioned work [16], where the analysis of boxers' strikes was carried out, the MLP configuration was as follows: "MLP-NN: Activation = tanh, alpha = 0.0001, hidden layer sizes = 8, 8, 8 (3 hidden layers with 8 nodes each), learning rate = constant, solver = lbfgs". In order for the neural network to learn correctly based on a given data sample with the greatest accuracy, it is necessary to select the optimal number of neurons. There should not be many of them, but there should also be enough for a good approximation of the function in the space of synaptic weights. If the number of neurons in the hidden layer is not high enough, then the problem of learning based on the training set cannot be solved. If there are too many neurons, the network may learn excessively [21].

The topology of the ANN (Figure 4) in this work was as follows:

- Input layer—600 nodes (600 values of absolute accelerations and angular velocities);
- Hidden layers—512, 256, 128, and 64 (four hidden layers);
- Output layer—4 nodes (straight punch, hook, uppercut, and movement without punches).

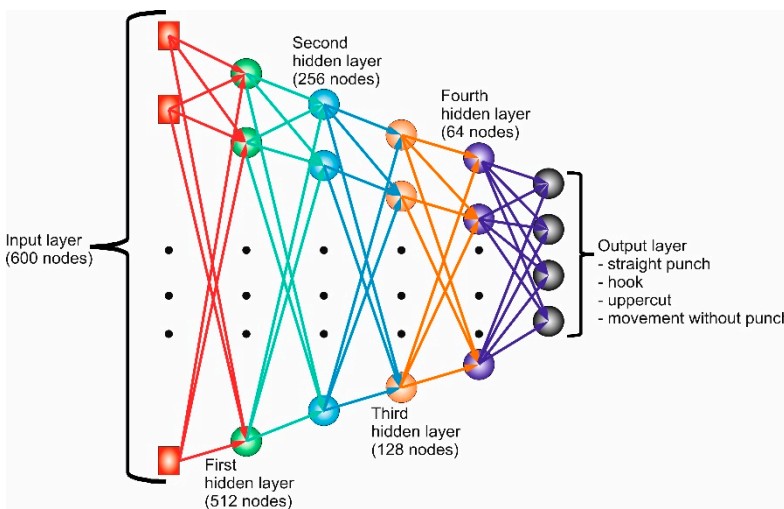

**Figure 4.** The topology of the artificial neural network (ANN).

In work [21], it states, "An MLP with one hidden layer has limited capacity, and using an MLP with multiple hidden layers can learn more complicated functions of the deep neural input. That is the idea behind deep neural networks where, starting from networks raw input, each hidden layer combines the values in its preceding layer and learns more complicated functions of the input".

Data processing was carried out using the Keras library. Keras is a deep learning API written in Python, running on top of the machine-learning platform TensorFlow. It was developed with a focus on enabling fast experimentation [20].

The activation function determines the output value of the neuron, depending on the result of the weighted sum of the inputs and the threshold value. In our work, the

sigmoid activation function was applied to each layer. The sigmoid activation function (1) is as follows:

$$\text{Sigmoid}(x) = \frac{1}{1 + e^{-x'}} \tag{1}$$

where x is the input tensor.

When training the ANN, the loss function was minimized, which, when using the Keras library, was specified as a parameter of the compile method (sets up the model for training) of the model class [20]. Since the data were categorical in our case, a loss function of the form sparse categorical crossentropy was used.

The Adam optimization algorithm was used for the model. Adam optimization is a stochastic gradient descent method that is based on an adaptive estimation of first-order and second-order moments [20].

During the training process, the network scans the training sample in a certain order. One complete pass through the sample is called a learning epoch. In total, 100 epochs were applied to train the models.

The classification included four categories:

- Straight punch (jab or cross);
- Hook;
- Uppercut;
- movement without punches.

The category of "movement without punches" was introduced, since, during movement, boxers often make rather sharp hand movements and they had to be distinguished from the actual punches.

After MLP training, the model was saved in a file with the .h5 extension, also using the Keras API, while the network architecture and weights were saved together in one file. Then, the model from the file was converted into the code of the stm32f407 microcontroller using the X-CUBE-AI code generator (Figure 5) [17].

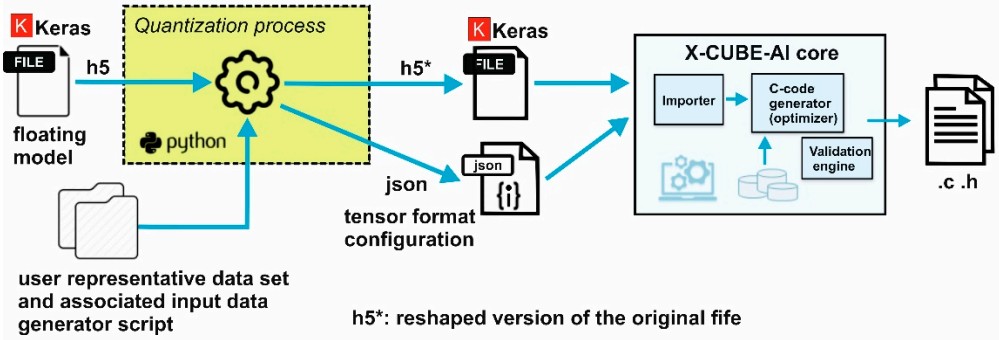

**Figure 5.** Generation and implementation of the pre-quantized Keras model.

The code generator quantizes weights and bias and associated activations from the floating point to an 8-bit precision. These are mapped onto the optimized and specialized C implementation for the supported kernels [17].

### 3.4. Experimental Procedure

The experiment involved three categories of volunteer boxers from the sports club of the Boxing Federation of the Republic of Tatarstan (Kazan, Russia): Group 1—this was the initial group, including boxers in their first year of training, and a total of 35 people (25 people—training data and 10 people—evaluation data); Group 2—2–3 years of study, 35 people (25 people—training data and 10 people—evaluation data); and Group 3—from 5 years of study, 15 people (10 people—training data and 5 people—evaluation data). All boxers were male and in all groups, their age ranged from 18 to 27 years old. The main part of the group had to participate in the dataset for training the model, whilst the control

part took part in evaluating the trained model. The study was conducted in accordance with the Declaration of Helsinki, and the protocol was approved by the Ethics Committee of 20200901a.

To collect data for training the ANN, each boxer applied single punches for each of the categories (straight, hook, and uppercut), and performed movements, imitating shadow boxing without punches. The punches were not differentiated by which hand they were punched with: The right hand or the left. Each boxer from each group performed 100 punches with each arm, followed by a rest period.

This series took 7 training days with breaks (one training day and one rest day) and each boxer conducted 6000 punches with both hands in each category. These data were used to train the model. A total of 1000 punches were used to form a test dataset that was used to validate the model.

These data were then used to train the ANN for each category of punches and each group of boxers. The ANN was also trained, in which data from all groups of boxers were used, after which all of these trained models were "frozen".

The model was stored in the microcontroller so that, in the second series of experiments, punch recognition occurred in the microcontroller. The process of recognizing punches in the microcontroller program was performed within 3 milliseconds, which is not a very fast process, but since the punch could last 100 times longer, the recognition occurred on-the-fly.

We implemented a control series, which consisted of 1000 punches with both hands in each category. This series of experiments involved the control parts of each group, that is, those who did not participate in training the model. The series was applied for the verification of trained models.

The models used for verification were those that were trained for each individual group and universal, and were trained based on data from all groups.

## 4. Results and Discussion

After producing a complete dataset, the models were trained for each group of boxers. MLP training took no more than 10 min on a computer with the following parameters: CPU—Intel Core i7; RAM—16 GB; and CPU frequency—3.2 GHz.

To assess the effectiveness of the models, the loss and accuracy functions were used. Figures 6–9 present graphs of the loss function and the accuracy function for each group of athletes. On the graphs, "Train" is the main dataset, which was used to train the model, and "Test" is the test dataset, which was used to validate the model.

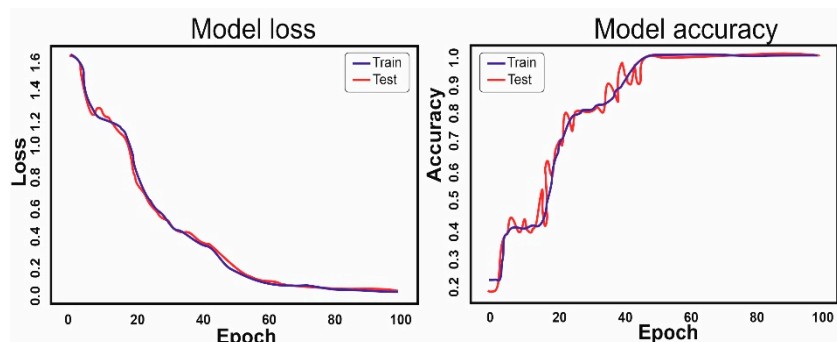

**Figure 6.** Graphs of the loss function and accuracy of boxers with 1 year of training experience.

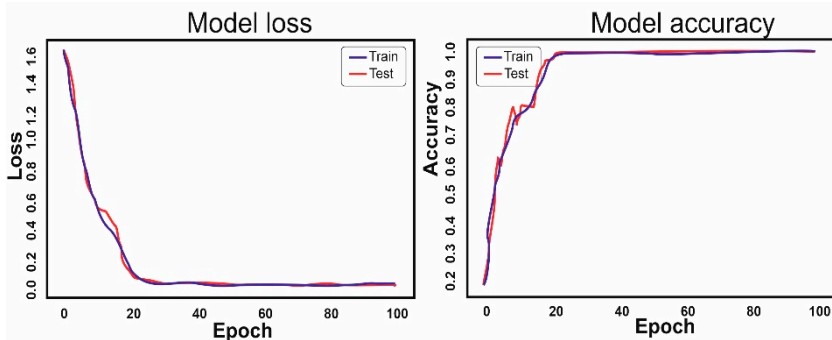

**Figure 7.** Graphs of the loss function and accuracy of boxers with 2–3 years of training experience.

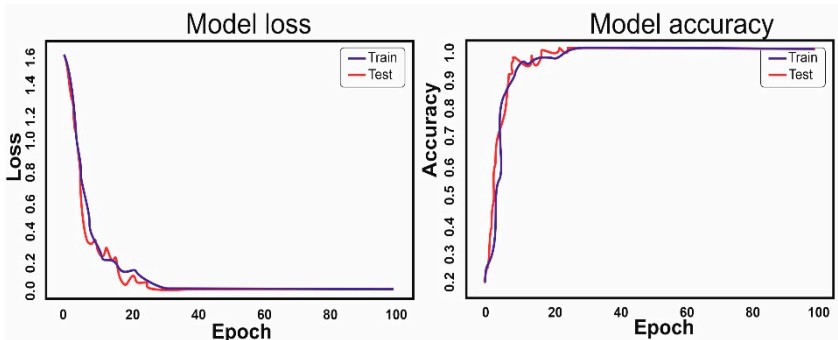

**Figure 8.** Graphs of the loss function and accuracy of boxers with more than 5 years of training experience.

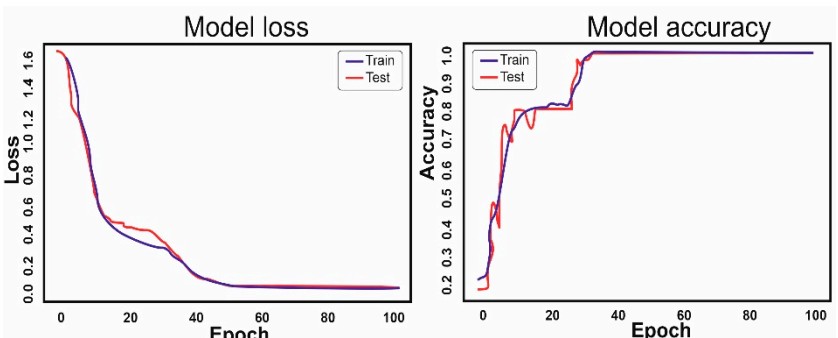

**Figure 9.** Graphs of the loss function and accuracy when training a universal ANN.

The analysis of the graphs shows that by the 60th epoch, the accuracy approaches 1. It is interesting that with the same training parameters among boxers with 1 year of training experience, outliers of the test and basic data divergence are clearly visible on the graphs, and are much greater than on the graphs of more experienced boxers.

The graphs show that the network settings are suitable for this type and amount of data. The paper [16] states that "all models reach a validation score of 1.0 at a training size of approximately 100". Moreover, in [16], MLP does not show the best dynamics of the learning process. It can be assumed that in [16], there was a significantly smaller dataset, compared to our work. This is important for MLP. In addition, it is not clear from the work [16] what parameters were at input to the MLP. The authors point out that the MLP had three hidden layers of eight nodes, and the measurements were carried out using an IMU that contained an accelerometer, gyroscope, and magnetometer. The input must contain data from three devices, and the punch lasts from 100 to 300 milliseconds. However, in general, we can say that the data obtained by us are comparable with the work [16].

In this series, the analysis of punches and recognition of the type of punches occurred for groups that did not participate in the model training series. No differentiation was

made by the types of punches; the general level of recognition was determined for all punches. The recognition level was calculated as follows:

$$\text{Recognition level} = \frac{\text{number of correctly recognized punches}}{\text{number of incorrectly recognized punches}} \times 100\%. \tag{2}$$

In the first group of boxers (1 year of training), recognition averaged $87.2 \pm 5.4\%$ (see Figure 10, Group model). It was difficult to predict such a situation in advance, but the level of punch recognition according to the universal model turned out to be higher—$91.89 \pm 3.45\%$ (Figure 10, Universal model).

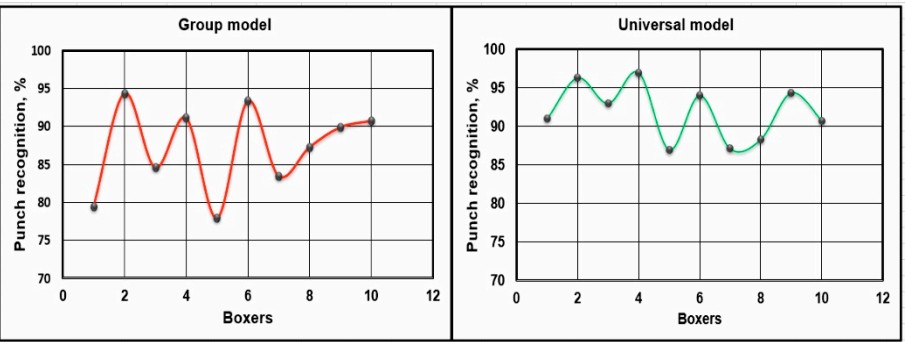

**Figure 10.** Graphs of the level of recognition of punches of a group of entry-level boxers.

The level of recognition is quite high, but from the graph, it is easy to see a large scatter of values, which can be explained by the fact that the technique of punches in this category of athletes does not yet have a constant basis.

Figure 11 shows a graph of the experiment of the 2nd group of boxers. The punch recognition rate is $95.33 \pm 2.51\%$ (Figure 11, Group model). This is the highest result of all three groups. The recognition of punches according to the universal model did not practically change and amounted to $94.17 \pm 2.96\%$ (Figure 11, Universal model).

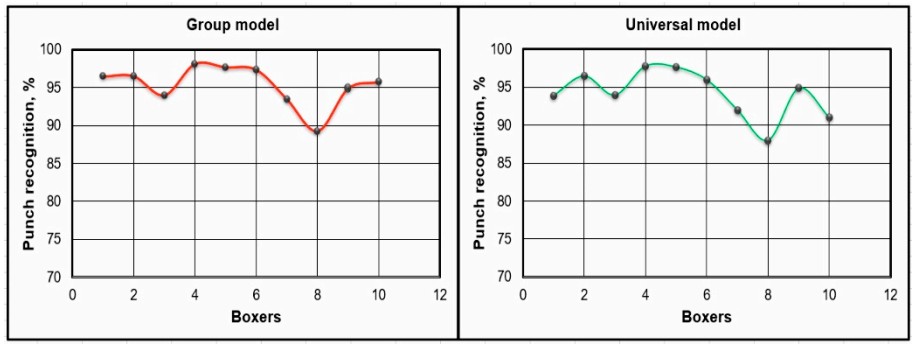

**Figure 11.** Graphs showing the level of recognition of punches of a group of boxers with 2–3 years of training experience.

Figure 12 shows a graph of the experiment of the 3rd group of boxers. The punch recognition level is $91.73 \pm 3.82\%$ (Figure 12, Group model). The lower level of punch recognition and large spread of values can be explained by the fact that high-level athletes have their own peculiar technique of punching. According to the universal model, as well as for group 1, the level of punch recognition was slightly higher—$92.93 \pm 4.33\%$ (Figure 12, Universal model).

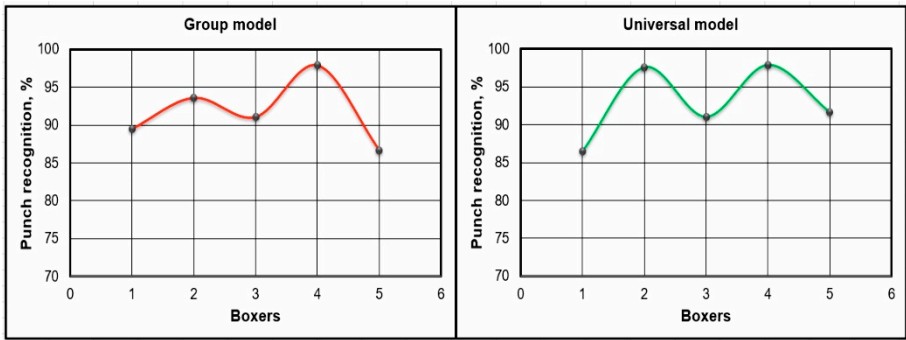

**Figure 12.** Graphs showing the level of recognition of punches of a group of advanced boxers (more than 5 years of training).

The obtained data are in good agreement with the data of other authors. In work [3], where nine people were used for the dataset, and 20% of them were the control group, they only participated to determine the accuracy of the model. The recognition accuracy varied for different types of movement from 87% to 99%.

In [11], based on video data analysis, Multi-class SVM classifiers and a Breiman Random Forest are utilized for punch classification. The experimental results illustrate an accuracy of 97.3% for previously seen athletes and 96.2% for unseen athletes. It should be noted that, in our work, the accuracy of the previously seen group of boxers was 100%.

In [22], based on video motion capture using Convolutional Neural Networks with a Long Short-term Memory Network, the accuracy of motion recognition was up to 78%.

The classification of karate strikes in a study by the authors of [12] based on video capture of movements showed a strong difference in accuracy, depending on the type of movement, ranging from $81 \pm 15\%$ to $97 \pm 4\%$.

In [16], one person participated in the study. This work used data that were obtained from IMUs. These data were used to recognize punches employing a variety of deep learning methods, including MLP. For the experiment configuration, when IMUs were attached to the boxer's wrists, the accuracy was 0.98. In work [16], configuration 2 was also investigated, in which an additional IMU sensor was installed in the third thoracic vertebrae region. The authors obtained an interesting result: "sensor 1 configuration demonstrated greater overall classification accuracy for all models".

## 5. Conclusions

Experiments have shown that the use of ANN in the form of MLP greatly facilitates the collection of data on the kinetic characteristics of boxers' punches, allowing this process to be automated. Parameters of punches, such as the speed and acceleration, can be determined during shadow boxing, while the type of punch and its parameters can be identified. It is known that the characteristics of a punch in a fight and when a boxer strikes outside of a fight are different. It is not difficult to measure the parameters of a punch during a fight with the modern development of technology, but to identify punches, one needs to specially conduct video capture of movements, and this is a very time-consuming and expensive procedure, which is practically very difficult to carry out for several dozen boxers. However, in the future, with the further development of technologies, video capture may take a leading role in recognizing movements in martial sports.

Machine learning often uses a data augmentation strategy that allows one to significantly increase the diversity of data available for training models. In this work, the method of "repetition without repetition" was used; that is, boxers were instructed to apply each punch differently, without changing the type of punch. At the same time, data augmentation was applied, which increased the efficiency of the developed model.

Separate models were developed based on data for specific groups of boxers, divided by the level of training. A universal model was also developed, based on data

from all groups. At the same time, the use of the universal model showed a better classification accuracy.

This makes it possible to put forward the hypothesis that it is possible to develop an ideal model of a boxing punch, based on the data of highly qualified boxers. Furthermore, the level of deviation from this model will correlate with the level of development of the athlete's boxing technique.

This hypothesis is supported by the fact that the worst results in terms of the punch recognition accuracy were shown by the group of boxers with the initial level of training (for the universal model, $91.89 \pm 3.45\%$) and the group of high-level boxers (for the universal model, $92.93 \pm 4.33\%$). It can be concluded that this is due to the fact that these boxers' punches differ too much in technique.

**Funding:** This research received no external funding.

**Institutional Review Board Statement:** The study was conducted in accordance with the Declaration of Helsinki, and the protocol was approved by the Ethics Committee at Financial University under the Government of the Russian Federation (20200901a).

**Informed Consent Statement:** Informed consent was obtained from all subjects involved in the study.

**Data Availability Statement:** Data available on request due to privacy restrictions.

**Acknowledgments:** The author would like to thank athletes, coaches, and the leadership of the Boxing Federation of the Republic of Tatarstan (Kazan, Russia) for organizing the experiments and actively participating in the study.

**Conflicts of Interest:** The authors declare no conflict of interest.

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
