# Peer review of "Application of an Artificial Neural Network to Automate the Measurement of Kinematic Characteristics of Punches in Boxing"

_applsci, doi:10.3390/app11031223_

Round 1

Reviewer 1 Report

see attachment

Author Response

Thank you for the good comments that allowed us to improve the quality of the article. I have revised many of the provisions of the article very much.

Reviewer 2 Report

Brief Summary:

The aim of the research was to implement an automated analysis system to determine shadow boxing punch types from different athlete levels of boxing experience using inertial measurement units. The author used a neural network with a multi-layer perceptron to classify the punches which resulted in fairly high accuracy rate compared to similar research. Ultimately, this research qualifies as a progressive addition to the literature in that it provides guidance for the application of wearable technology in combat sports. Generally, as the current research is inclined and should be directed towards an applied context in sport rather than a theoretical paper, there is scope here to further cater to this and expand on how your research can be applied practically for coaches, sport scientists, sport medical staff and athletes.

Overall, the author has designed and undertaken a well-researched study that responds to previous literature gained knowledge and the current trends of innovation in sport performance analysis.

Broad Comments: 

  • A full review of the abstract is recommended for sentence structure, grammar and flow. Furthermore, it is worth stating that you a specifically looking at shadow boxing here as per the main aim in the introduction. Also, please finish the abstract with key research result outcome and practical take-home message of the study, not just results.  
  • Shadowboxing is the sport movement being recorded; how does this translate to real boxing? The data signal properties would be very different with actual impacts being recorded which as well might influence the analysis method used. Do you think your research translates to actual boxing whether that be in training or competitions? Of are you specifically targeting the training drills of shadow boxing? Do you think that if used when boxers are striking an opponent or bag that vision may be a data source to aid in punch type recognition?
  • The accelerometer range is stated as ±16g, did you experience any data saturation? Do you think this range would be suitable for when impact punches are performed in terms of capturing the whole punch signal without clipping?
  • The discussion and conclusion are very brief and do not apply the theoretical developments of the paper into a practical aspect for sport. Could you expand on how your system and capturing process can be applied to individual athlete performance enhancement? Could you specify if this is a developmental concept and if so what progressions in methods or technology are needed to apply this concept practically?

Specific Comments:

Abstract

  • Line 8: worth specifying what you mean by “speed qualities of boxing”, are these punch, athlete physical movements like change of direction.
  • Line 15: insert “the” Infront of ANN.
  • Line 15: these are derived parameters from IMU data, not raw IMU data, please revise how you have written this sentence.

1: Introduction

  • Lines 25 – 27: These sentences can be combined and information conveyed made more concise.
  • Lines 42 – 45: This is a very long and drawn out sentence, please revise to make more concise.
  • Lines 56 – 57: why is artificial neural networks listed separately here? Would you not consider them an algorithmic method for Machine or Deep Learning?
  • Line 58: a subjective and throw away sentence – either refine and reference or remove.

2: Related works

  • Lines 99 – 101: This section seems a direct copy of the information in the last part of the Introduction, please remove one of these to avoid unnecessary duplication.
  • Lines 102 – 135: There is minimal writing flow in this section and reads more like a list. Can you re-write this section and expand more on how the cited research is in context to your research study and how your collated literature highlights the research gap you are targeting?  

3: Materials and Methods

3.1 Design of experiment

  • Line 138: Be consistent with the terminology used for naming the inertial units.
  • Lines 142 -143: What was the threshold required to record a punch?

3.2 Measuring modules

  • What is the weight of the IMU device?
  • Lines 155-158: what was the sampling rate in Hertz of the IMUs?
  • Lines 169 – 171: Could you expand on the actual punch detection steps that recorded and segmented punches from the raw data?

3.3 Neural network architecture

  • Why was a neural network only chosen? Were other algorithms tested prior as a comparison? If so, can you report these?
  • The two data input parameters were absolute accelerations and angular velocities, what was the decision behind not using or testing data of individual axes or other derived features?
  • Was any data feature engineering or selection methods used to improve model input? If not, why? Are you able to report on the accuracy vs feature set size using different feature subset selections?
  • Lines 169 – 170: This is a disjointed sentence, please split or edit for clarity.

3.4 Experimental procedure

  • Lines 198 – 202: This is very confusing and hard to read how it is currently written. Please revise for clarity of the participant information.
  • Lines 212 – 218: This contextual information is good background but better placed in the introduction or further expanded on in the discussion as it is not an actual method procedure of your research.
  • What were the model computation times and hardware operating system specifications?
  • What is the computational complexity in terms of mean training time for your models? Will this allow for potential real-time feedback to the athletes in training?

4: Results

  • Model accuracy and loss functions are used to evaluate the models, why are no further evaluations made for example F1- score common in similar research for sport-specific movement recognition to allow for comparisons and context for readers?
  • Figures 4 -7: Move the legends so they are not covering the actual data points.
  • Line 261: Which graph? Reference what you are referring to.
  • You used accelerometer and gyroscope data and obtained accuracies between 87-95%, can you explain how would the accuracy have improved if you were to have used sensor fusion data in terms on Euler orientation angles (roll, yaw and pitch)? These angles can be obtained from a combination of accelerometer and gyroscope (sometimes magnetometer too) and it's currently a very common practice.

5: Discussion and conclusions

  • Lines 286 -303: rather than re-stating and duplicating the results, could you expand on discussing the differences between the groups, why this may occur, how this compares to other sport-specific movement recognition studies using IMUs, implications for using your system in boxing training or live?
  • Line 311: What experiments? Reference what you are referring to.
  • You acknowledge that punching in fights would lead to different data characteristics as to training punches, can you speak to this more in terms of your study methods and findings?
  • Yes, the process of training a computer vision system for sport movement recognition can be time-consuming although there are ever-developing methods to speed this process up. Could you comment on the trade-offs between using either IMUs and/or vision processes from experimental set-up to model evaluation, for example, the balance between model computational efficiency, results accuracy and complexity trade-offs calculations?

Author Response

I want to thank you for your valuable advice and comments. They allowed me not only to improve the article, but also to gain a deeper understanding of the research topic.

Reviewer 3 Report

In this manuscript, the author shows the study of the automation of measuring the speed qualities of boxers using inertial measurement units (IMUs) based on an artificial neural network. Besides, the author shows experiments with different levels of training. 

The work is interesting and tackles an up-to-date topic. However, I think the author should illustrate the innovative point with most existing algorithms. Below are my two major concerns on this work:

(1) The author shows the results of the experiments, but do not give a theoretical analysis to support these results, which makes that the contribution is vague, deficient and the core idea is not supportable.

(2) The work is not experimentally compared to any existing approach, which again minimizes the contribution. Moreover, the tool(s) and parameters used for simulation are vague. Therefore, the realism, credibility, and appropriateness of the shown simulations are questionable.

Author Response

The work is interesting and tackles an up-to-date topic. However, I think the author should illustrate the innovative point with most existing algorithms. Below are my two major concerns on this work:

From my review of the literature in this area, only one article uses more than one algorithm.

(1) The author shows the results of the experiments, but do not give a theoretical analysis to support these results, which makes that the contribution is vague, deficient and the core idea is not supportable.

The article describes in detail:

  • the type and technical characteristics of the equipment;
  • the type and parameters of the artificial neural network.

All the articles that you can see in my help rewrite the theoretical justification from other articles and books, if this is the justification, then I supplemented the article with them.

All articles research from 1 to 10 people who made 20 to 100 punches, and the authors base their conclusions on this. I examined 85 people, each of whom made 1000 to 7000 punches. The theory shows a pattern that should be statistically reliable.

(2) The work is not experimentally compared to any existing approach, which again minimizes the contribution. Moreover, the tool(s) and parameters used for simulation are vague. Therefore, the realism, credibility, and appropriateness of the shown simulations are questionable.

The criterion of truth is practice, and the basis of science is the reproducibility of results. Using this data, you can easily reproduce ANN:

The topology of the ANN in this work was as follows:

  • - input layer – 600 nodes (600 values ​​of absolute accelerations and angular velocities);
  • - hidden layers – 512, 256, 128, 64 (4 hidden layers);
  • output layer – 4 nodes (straight punch, hook, uppercut, movement without punches).

The analysis of the graphs shows that by the 60th epoch, the accuracy approaches 1. It is interesting that with the same training parameters among boxers with 1 year training experience, outliers of the test and basic data divergence are clearly visible on the graphs, which are much larger than on the graphs of more experienced boxers.

The graphs show that the network settings are suitable for this type and amount of data. The paper [17] states that "all models reach a validation score of 1.0 at a training size of approximately 100". Moreover, in [17] MLP shows not the best dynamics of the learning process. It can be assumed that in [17] there was a significantly smaller dataset, compared to our work. This is important for MLP. In addition, it is not clear from the work [17] what parameters were at the input to the MLP. The authors point out that the MLP had three hidden layers of eight nodes, and the measurements were carried out using an IMU that contained an accelerometer, gyroscope, and magnetometer. That is, the input must contain data from three devices, and the punch lasts from 100 to 300 milliseconds. But in general, we can say that the data obtained by us are comparable with the work [17].

The obtained data are in good agreement with the data of other authors. In work [3], where 9 people were used for the dataset, and 20% of them were the control group, that is, they participated only to determine the accuracy of the model. The recognition accuracy varied for different types of movements from 87 to 99%.

In [11], based on video data analysis and Multi-class SVM classifiers and a Breiman Random Forest are utilized for punch classification. Experimental results illustrate the accuracy of 97.3% on previously seen athletes, and 96.2% on unseen athletes. It should be noted that in our work, the accuracy of the previously seen group of boxers was 100%.

In [23], based on video motion capture using Convolutional Neural Networks with a Long Short-term Memory Network, the accuracy of motion recognition was up to 78%.

The classification of karate strikes in a study by the authors of [12] based on video capture of movements showed a strong difference in accuracy depending on the type of movement: from 81% ± 15% to 97% ± 4%.

In [17], one person participated in the dataset set. This work used data that was obtained from IMUs. This data was used to recognize punches using a variety of deep learning methods, including MLP. For the experiment configuration, when IMUs were attached to the boxer's wrists, the accuracy was 0.98. In work [17], configuration 2 was also investigated, in which an additional sensor IMUs was installed in the third thoracic vertebrae region. The authors got an interesting result:"sensor 1 configuration demonstrated greater overall classification accuracy for all models“

Round 2

Reviewer 3 Report

The authors replied to all my concerns, I have no further comments.